# Hot Corrosion Behavior of TWAS and HVOF NiCr-Based Coatings in Molten Salt

**DOI:** 10.3390/ma16041712

**Published:** 2023-02-18

**Authors:** Kateřina Lencová, Marie Frank Netrvalová, Marek Vostřák, František Lukáč, Radek Mušálek, Zdeněk Česánek, Šárka Houdková

**Affiliations:** 1Research and Testing Institute Plzen, 30100 Pilsen, Czech Republic; 2Institute of Plasma Physics of CAS, 18200 Prague, Czech Republic

**Keywords:** hot corrosion, molten salt, NiCr-based coating, HVOF, TWAS, X-ray diffraction, gravimetric method, Raman spectroscopy

## Abstract

In order to extend the life of boilers by applying an anti-corrosion coating without the need to dismantle them, it is advisable to find coatings that can be applied using cheaper and portable techniques, such as Twin Wire Arc Spray technology (TWAS). In this study, we compare selected NiCr-based coatings and two uncoated steel substrates (steel 1.7715 and 1.4903). Two coatings, Cr_3_C_2_ - 25% NiCr and Hastelloy C-276 are deposited using High velocity oxygen-fuel technology (HVOF) and three coatings, NiCrTi, NiCrMo, and Inconel 625, are deposited using TWAS. In addition to the corrosion weight gain during 50 cycles of loading in an 18% Na_2_SO_4_ and 82% Fe_2_(SO_4_)_3_ salt environment at 690 °C evaluated using the gravimetric method, the microstructure and phase composition of the coatings were analyzed on the samples after the exposure in order to compare the properties and gain a deeper understanding of the corrosion kinetics. Coating cross-sections and free-surfaces were observed with a scanning electron microscope (SEM) with an energy-dispersive (EDX) system. The phase composition was investigated using X-ray diffraction (XRD) and Raman spectroscopy. No significant differences were observed between the TWAS and HVOF coating methods for the coatings compared. Due to the similar corrosion products found on all coatings, a very effective corrosion protective layer was formed on the surface, forming a barrier between the corrosive environment and the coating regardless of the used deposition technology. Therefore, for industrial use on the inner surface of coal-fired boilers we recommend NiCrTi, NiCrMo, or Inconel coatings prepared with the more cost-effective and portable TWAS technology.

## 1. Introduction

The protective thermal spray coating applied to the surface of solid fuel boiler components is designed to ensure operation under extreme conditions and extend service life by creating a barrier between the corrosive environment and the substrate material, thereby providing a greater resistance to corrosion and oxidation [1,2,3].

In an effort to increase the efficiency of solid fuel boilers their operating temperatures are being increased. However, the operating temperatures of the boilers are limited by the corrosion resistance of the component materials. High-temperature corrosion of the superheater and preheater tubes is a serious problem that leads to a gradual loss of tube wall thickness. No alloy is immune to high-temperature corrosion for an indefinite period, although some alloy compositions have a long initiation time before the actual corrosion process occurs. These alloys, named superalloys, which are designed for high-temperature applications are, however, unable to meet both the sufficient thermal strength and resistance to high-temperature corrosion. For this reason, protective thermal spray coatings are applied to high-temperature equipment [1,4,5,6,7]. High-temperature corrosion is an accelerated corrosion in which salts are deposited on the surface of the component material. These low melting point salt mixtures attack the protective oxide layer of the material and accelerate corrosion. High-temperature corrosion generally occurs, for example, in the materials of boilers, gas turbines, internal combustion engines, and industrial waste incinerators [1,4,5,6,7]. Due to the depletion of a high proportion of the world’s fossil fuel reserves, relatively low-quality fuels are being burned. Such fuel (coal) then contains a high percentage of impurities such as sulphur, sodium, or potassium. High-temperature corrosion in solid fuel boilers is related to these impurities, which are present in the ash formed during combustion. The fly ash settles on the surface of boiler components (e.g., boiler walls, superheaters, preheaters, etc.) and forms complex compounds with a low melting point that cause high-temperature corrosion. The typical corrosive environment for solid fuel boilers is 18% Na_2_SO_4_ and 82% Fe_2_(SO_4_)_3_ [7,8,9,10]. High-temperature corrosion in this environment is addressed, e.g., in studies [8,10,11,12,13].

Due to the uneconomical aspect of dismantling the boiler to apply a protective anti-corrosion coating, it is more than advisable to look for options with portable on-site coating techniques such as Twin Wire Arc Spray technology (TWAS), which recently became readily available as an alternative method to more traditional High Velocity Oxygen-fuel Technology (HVOF) spraying, which usually demands off-site spraying in a dedicated spray shop [14,15]. Although some properties are worse due to the use of TWAS technology [16,17], other studies indicate that they may not achieve worse results in a particular environment [18].

In this paper, the focus is on finding a suitable NiCr-based coating material concerning corrosion kinetics in a combustion boiler environment simulated by the cyclic loading of a sulphate molten salt environment at 690 °C. NiCr-based alloys or high chromium alloys are often used as coatings to resist high-temperature corrosion. Dense protective oxides of chromium, nickel, and their spinels create a diffusion barrier to inward oxidation, thereby increasing corrosion protection [3]. Low-melting eutectic salt Na_2_SO_4_ and 82% Fe_2_(SO_4_)_3_ represent an environment usually found in coal-fired boilers where the fly coal ash deposit onto boiler tubes and induce high-temperature corrosion. 

Two uncoated materials (common steel used in coal-fired boilers 1.7715 and expensive corrosion-resistant steel 1.4903) and NiCr-based coatings prepared using two methods were selected for the comparison of corrosion kinetics, microstructure, and phase composition. The first method is HVOF, which, due to the high velocity of the powder particles on impact, produces a coating with a more uniform dispersion and anchoring of the particles to the substrate material. The second method is TWAS, which is expected to have higher porosity due to its nature but in practice is not only cheaper but also more portable. In addition to the corrosion weight gain under 50 cycles of loading in a salt environment of 18% Na_2_SO_4_ and 82% Fe_2_(SO_4_)_3_ at 690 °C, the microstructure and phase composition of the coating were analyzed after exposure for property comparison and a deeper understanding of the corrosion kinetics.

## 2. Materials and Methods

### 2.1. Materials

Several commercially available NiCr-based coatings deposited using HVOF and TWAS as well as pure substrates were evaluated in this experiment. Coatings were applied using HP/HVOF technology with a JP/5000 torch (TAFA Inc., Miami, FL, USA) and a TWAS SmartArc (Oerlicon Metco, Pfäffikon, Switzerland) spraying system on specimens from a high-temperature corrosion resistant chromium-molybdenum steel 1.4903 (X10CrMoVNb91). The thickness of all coatings was 400 ± 50 μm. To observe the behavior of the uncoated material in a corrosive environment, specimens from 1.7715 ordinary steel commonly used in coal-fired boilers and also 1.4923 were tested. The specimens’ sizes were 20 mm × 20 mm × 5 mm. 

A detailed overview of the studied coatings is in Table 1. The chemical element composition of the coatings (see Table 2) and steels (see Table 3) were taken from the supplier data sheets except the coating marked as NiCrMo, on which a separate chemical analysis was performed. As this is a cored wire, the composition of the core and the sheath are reported separately. The cored wire was cut and embedded in epoxy and an EDS analysis was performed on the cut at specific points. 

### 2.2. Methods

The laboratory high-temperature corrosion test was designed to simulate the real-world environment of power-generation systems. The specimens were exposed to a molten salt environment of 18% Na_2_SO_4_ + 82% Fe_2_(SO_4_)_3_ at 690 °C under cyclic conditions, simulating the working environments in the low-emission boilers of coal-fired power plants. The exposed surfaces of the specimens were polished down to Ra = 1 µm. A salt deposit (3–5 mg/cm^2^) of 18% Na_2_SO_4_ + 82% Fe_2_(SO_4_)_3_ with uniform thickness was applied with a brush on the samples preheated to 250 °C. Then the specimens were placed in a ceramic crucible. Each of the 50 test cycles included 1 h of heating at 690 °C in a furnace followed by 20 min of cooling at room temperature. Weight-change measurements were performed after each testing cycle to determine the kinetics of corrosion. The samples were weighed with the ceramic crucible, including material that spalled from the sample during the test.

Cross-sections of samples were observed using the EVO MA 15 (Carl Zeiss, Jena, Germany) scanning electron microscope (SEM) equipped with a XFlash 5010 energy-dispersive (EDX) system (Bruker, Heidelberg, Germany).

The coatings’ phase compositions were evaluated using X-ray diffraction (XRD) and Raman spectroscopy. For the XRD measurements, the D8 Discover powder diffractometer, Bruker, in Bragg-Brentano geometry with a 1D detector and CoKα radiation was used. The scanned region was from 15 to 100° 2θ with a 0.03° 2θ step size and with a 96 s counting time per step. The obtained diffraction patterns were subjected to quantitative Rietveld analysis [19] performed in the TOPAS 5, which uses the so-called fundamental parameters approach [20]. Crystalline phases in the XRD patterns were identified using the PDF4+ database and their unit cells were used for the whole diffraction pattern fitting in order to eliminate the peak overlap. The Raman spectroscopic analyses were performed using a DXR Raman microscope by means of backscattering geometry with a green (532 nm) Nd:YVO4 DPSS laser source. The spectra were recorded in 5 different positions in the range of 50–3500 cm^−1^ with a microscope magnification of 10× which provides a laser spot with a diameter of 2.1 µm.

## 3. Results

### 3.1. Corrosion Kinetics

The results of the test of resistance to high-temperature corrosion in an aggressive environment of sodium sulfate and ferrous sulfate salts were evaluated based on the thermogravimetric method of corrosion kinetics. The weight gains of all the evaluated coatings and pure steels are shown in Figure 1.

As shown in Figure 1, 50 cycles of the corrosion process, the weight gain of all coatings, and the corrosion resistant chromium-molybdenum steel 1.4923 showed very similar progress, with the final gain under 2 mg/cm^2^, in comparison to the ordinary steel 1.7715 showing approx. 30 times higher gain. In Figure 1, the results are presented each fourth point for better orientation and clarity.

During the first 10 cycles the process stabilized to where the values oscillated around zero and were therefore excluded from the evaluation. For steel 1.4923 and all coatings selected for this study, the increase in weight gain after 50 load cycles is very small and the corrosion rate of different coatings prepared using the two different techniques is comparable.

### 3.2. SEM/EDX Cross-Section Analysis

Figure 2 shows the results of the SEM analysis of the as-sprayed condition and after exposure to the corrosive environment. In the case of the TWAS coatings, the characteristic microstructure can be seen with a higher content of oxides which are already formed during the spraying process. On the surface of all the tested coatings is a visible surface corrosion attack with the formation of an oxide scale. The thickest oxide layer of all the tested coatings was observed with the Cr_3_C_2_ - 25% NiCr coating applied using HVOF technology with a thickness over 40 µm. On the surface of the bare steel a thicker oxide layer is formed than in the case of the samples with the thermal sprayed coatings. The oxide layer of steel 1.7715 tends to the spallation. 

The EDX analysis results, shown in Figure 3, Figure 4, Figure 5 and Figure 6, for one HVOF coating and one TWAS coating in the as-sprayed state and after exposure were selected for demonstration. 

The EDX maps in Figure 3 show the distribution of Ni-Cr-C elements in the as-sprayed state of the NiCr coating in the carbide matrix. Figure 4 describes the effect of the corrosive environment on the Cr_3_C_2_ - 25%NiCr coating: chromium has the highest abundance in the coating and a relatively homogeneous layer is formed at the interface with uniform chromium and oxygen content with very little to no carbon and nickel content. At the surface there is a portion of the coating with even lower chromium and oxygen content, but small amounts of carbon and nickel are present. The sodium map shows that sodium from the corrosion medium has not entered the corrosion layer or the coating and has not accumulated anywhere, whereas sulphur from the corrosion medium is present to a large extent at the corrosion layer/coating interface. There is more oxygen at the bottom of the corrosion layer and the iron is near the surface of the corrosion layer and is inhomogeneously distributed.

In the case of the NiCrTi coating prepared using the TWAS method, the gaps between the splats can be seen in the SEM image. The EDX maps show inhomogeneous parts of the splats on which, even in the as-sprayed state, inhomogeneously distributed Ti regions, regions with lower Ni content, and oxidized regions are visible. Chromium is more or less homogeneously distributed throughout the observed area of the coating, as shown in Figure 5. Figure 6 again shows the effect of the high-temperature corrosive environment on the NiCrTi coating. It can be seen that the corrosion layer contains less nickel and titanium than the coating. In contrast, the corrosion layer is composed of a mixture of chromium, sulfur, oxygen, and iron. Due to the uniform distribution of sodium throughout the EDX map, we can say that sodium is of an absolutely negligible amount in the study area.

### 3.3. XRD Analyses

A phase composition evaluation using XRD analysis was carried out on the surface of the specimen. The penetration depth of the used radiation was approximately 10–30 μm depending on the angle of incidence of the radiation beam on the sample surface, which means that the quantitative analysis presented in Table 4 reflects the information from the surface up to 30 μm. Identification of the oxides and spinels in the layer was performed using the ICSD database with the results of the EDX chemical composition mapping analysis of a cross-section of the samples.

The XRD analyses results of the surface oxides formed on the coated samples indicated the formation of oxide spinel, Cr_2_O_3_, Fe_2_O_3_, and NiO. In the case of the Hastelloy C-276 coating, NiMo_4_ and m-FeMoO_4_ were also detected. For the NiCrMo coating, the XRD analysis also shows the formation of Ni and Fe_2_W phases. A residual corrosive environment in the form of Na_2_SO_4_ was detected in the case of the coatings NiCrTi, NiCrMo, Inconel 625, and bare sample steel 1.4923. The oxide layers of the bare steel samples are formed by hematite (Fe_2_O_3_) and magnetite (Fe_3_O_4_)—oxide spinel. 

### 3.4. Raman Spectroscopy

Due to the complexity of the structures and other influences such as the large deformation and stresses in the coating lattice, it is very difficult to identify the individual phases of the coating, especially to distinguish the individual oxide spinels from each other using XRD as well as Raman spectroscopy, which are mutually complementary. A more detailed knowledge of the composition of the surface oxide layer can help predict the rate of oxidation and its passivation properties.

A Raman analysis was performed only on the coatings tested in the corrosive environment. Steels were not evaluated. 

The theory for the spinel structure suggests that there should be five Raman-active vibrational modes [21]. As can be seen from Figure 7, the strongest line of most measurements is at the position around 700 cm^−1^. This position corresponds more closely to the spinel NiFe_2_O_4_ [22,23,24,25,26].

It is very interesting to compare the measured spectra of individual coatings in terms of structural uniformity or heterogeneity. While Cr_3_C_2_ - 25% NiCr and NiCrMo coatings exposed to a corrosive environment show only slight deviations in vibration ratios of the same structure (Figure 7a,d), Hastelloy C-276, NiCrTi, and Inconel 625 coatings show a double type of irregularly alternating structure, where out of five measured points there are always two points of one type and three points of another type, so it cannot be said that this is a random deviation of measurements (see Figure 7b,c,e).

Measurements using Raman spectroscopy confirm the presence of residual amounts of Na_2_SO_4_ (see Figure 7a,e—the main peak v1 (A1) mode at 992 cm^−1^ and visible v3 (F2) modes at 1101, 1131, and 1152 cm^−1^ [23,27], respectively). 

Table 5 represents corrosion products detected by Raman spectrometry.

## 4. Discussion

### 4.1. Individual Samples

A continual corrosion of the oxide layer on the surface of all coatings formed by oxide spinel, Cr_2_O_3_, Fe_2_O_3_, and NiO shows the tendency to act as efficient diffusion barriers for the inward penetration of the corrosive environment. In the majority of tested samples, the presence of an inhomogeneously distributed residual amount of corrosive environment in the form of Na_2_SO_4_ was detected utilizing SEM+EDS as well as XRD analysis and Raman spectroscopy confirms this result.

There are many review studies (e.g. [3]) summarizing the various factors (temperature, gas composition, composition of molten salts, alloy elements, and external stress) acting on the corrosion behavior of alloys, which conclude that the dominant effect in NiCr-based alloys is the formation of a dense protective Cr_2_O_3_ barrier. 

The corrosion test shows very similar results to the study of Sidhu et al. [52] under real conditions or by Chatha et al. [53] with the study of NiCr and Cr_3_C_2_-NiCr coatings in Na_2_SO_4_-60%V_2_O_5_ at 750 °C.

#### 4.1.1. HVOF Samples

##### Cr_3_C_2_ - 25% NiCr

In the case of the Cr_3_C_2_ - 25% NiCr coating, the corrosive environment caused the formation of an oxide layer consisting mainly of chromium and nickel in the form of the oxide spinels, NiO, and Cr_2_O_3_. The XRD analyses results of the Cr_3_C_2_ - 25% NiCr coating are found to be similar to those reported by Kaur [12] for the D-gun sprayed Cr_3_C_2_-25% NiCr coating in the molten salt environment of 18% Na_2_SO_4_-82% Fe_2_(SO_4_)_3_. The formation of Cr_2_O_3_, NiO, and NiCr_2_O_4_ surface oxides has also been reported by Sidhu [54,55] for the HVOF sprayed Cr_3_C_2_ - 25% NiCr in the molten salt environment of 40% Na_2_SO_4_-60% V_2_O_5_. A higher concentration of chromium is located at the bottom of the corrosive layer, as can be seen from the EDX mapping, see Figure 4, while nickel is primarily on the surface of the oxide layer. Iron in the form of Fe_2_O_3_ detected using XRD analysis (Table 3) on the surface of the corrosion layer is the result of high-temperature exposure to the corrosion medium. Raman spectroscopy confirms these results. Sulphur diffused through the oxide layer to the surface of the coating from a corrosive environment. 

##### Hastelloy C-276

The oxide layer of the Hastelloy C-276 coating is mainly composed of chromium, molybdenum, and nickel. The XRD analysis shows the presence of the oxide spinels, Cr_2_O_3_, and NiO. The formation of these phases is in agreement with those reported by Česánek [56] for the HVOF sprayed Hastelloy C-276 coating in a molten salt environment of 40% Na_2_SO_4_-60% V_2_O_5_. A chromium-depleted area has formed in the upper layer of the coating. Molybdenum was detected using XRD analysis in the form of two oxides of the monoclinic phase, NiMoO_4_, and FeMoO_4_, respectively. Nickel is present in higher concentrations on the surface of the oxide layer. A higher concentration of iron was also detected on the surface of the oxide layer, which is Fe_2_O_3_ according to an XRD analysis. Tungsten and iron were detected in smaller volumes throughout the corrosion layer. The iron can come from the coating itself as well as from the corrosive environment. 

#### 4.1.2. TWAS Samples

##### NiCrTi

The corrosion layer of the NiCrTi coating is formed of chromium, iron, and nickel oxides in the form of the oxide spinels, Cr_2_O_3_, Fe_2_O_3_, and NiO. Figure 6 displays the EDX elemental mappings of the NiCrTi coating. Detected XRD phases are consistent with those reported by Wang [57] for the TWAS spray NiCrTi coating in a molten salt environment of 90% Na_2_SO_4_-10% NaCl. The results are found to be also similar to those reported by Guo [58] for the TWAS sprayed NiCrTi in a molten salt environment of 70% Na_2_SO_4_-30% K_2_SO_4_. The iron comes from a corrosive environment. The presence of the suplhur, which came from a corrosive environment, was detected on the interface of the oxide scale and coating material. 

##### NiCrMo

A thin corrosion oxide layer on the surface of the coating is mainly formed of silicon and iron with a small amount of nickel and chromium. Oxide spinel, Cr_2_O_3_, Ni, NiO, Fe_2_O_3_, and Fe_2_W were detected on the surface of the coating using XRD analysis. Iron can be assumed to come from a corrosive environment. The SEM analysis results show numerous decohesion between the splats in the upper part of the coating. In the upper part of the coating, a corrosion attack is also visible between the splats where the corrosive environment penetrated the coating. In the EDX analysis, the spectral lines of molybdenum and sulphur overlap and it is therefore not possible to determine which element is involved, but Raman spectroscopy shows a strong Mo-O vibration. The splat boundaries, especially in the upper part of the coating, are formed by chromium oxides, which were not developed in the as-sprayed state. Aluminum based oxides are also visible at the splat boundaries.

##### Inconel 625

In the case of the Inconel 625 coating, the corrosion layer is composed of nickel, tantalum, and chromium with a nickel-depleted area underneath. There is a thin layer of iron on the surface of the corrosion layer which originates from a corrosive environment. The results of XRD analysis revealed the presence of NiO, oxide spinel, Cr_2_O_3_, and Fe_2_O_3_. The formation of NiO, NiCr_2_O_4_, and Cr_2_O_3_ phases has also been reported by Fesharaki [59] for laser and TIG cladded Inconel 625 coatings in a molten salt environment of 40% Na_2_SO_4_-60% V_2_O_5_. In the as-sprayed condition, chromium and niobium oxides were detected, which further developed at the spall boundaries after exposure to the corrosive environment. After exposure to a corrosive environment, a slight corrosion attack was observed at the interface between the splats in the upper layer of the coating. In the EDX analysis, the spectral lines of molybdenum and sulphur overlap and it is therefore not possible to determine which element is involved. Raman spectroscopy did not reveal any visible Mo-O bonds, so it can be assumed that, in this case, it is sulfur from a corrosive environment.

#### 4.1.3. Steel

##### Steel 1.4923

Steel 1.4923 is marked as stainless heat resisting chromium steel with a molybdenum additive. The corrosion scale formed on uncoated steel 1.4923 is mainly formed by iron and chromium. The corrosion consists of two layers; the upper layer consists of Fe_2_O_3_ and the bottom layer is formed by Fe_3_O_4_. Below the interface between the oxide layer and the substrate material a high concentration of Cr is located, below which is a depleted area. Sulfur from the corrosive environment diffused through the oxide layer.

##### Steel 1.7715

The SEM analysis results of the uncoated steel 1.7715 show a thick oxide layer with apparent layering and a tendency to flake off. The oxide layer is mainly composed of iron in the form of oxide spinel—in this case Fe_3_O_4_ and a small amount of Fe_2_O_3_, which indicates a non-protective condition. As this is a common steel used in the manufacturing of coal-fired boilers, and showed the worst corrosion resistance in the simulated environment, it is quite obvious that it is necessary to coat such steel. To compare the behaviour of uncoated common steels, stainless heat resisting chromium steel with molybdenum additive 1.4923 was also used, but due to its high cost it is not economical to use it.

### 4.2. Economical Aspects

The price per 1 kg of commercially available Ni-based additive materials, whether in the form of powder for HVOF or wire for TWAS, does not differ much from each other. Their total amount is rather determined by the current market price of individual components and the total amount of supplied material.

However, the deposition efficiency of HVOF is significantly lower compared to TWAS. Based on experience from industrial practice, the material consumption can be estimated to be up to three times higher for the HVOF spraying of the Cr_3_C_2_ - 25% NiCr material and up to six times higher with the Hastelloy C-276 material than for the spraying of Ni-based alloys using TWAS.

The time required for spraying and the related costs of operating the equipment also play a significant role. The spraying time of a coating with comparable area and thickness is up to three times longer in the case of HVOF technology. Energy costs and consumption of the working medium (kerosene, oxygen, and nitrogen in the case of HVOF, compressed air in the case of TWAS) are up to 20 times higher for HVOF.

The combination of the above aspects leads to a final cost 4–5× higher when using HVOF technology.

## 5. Conclusions

This study focused mainly on the search for a replacement for coatings prepared using the HVOF method. Coatings prepared using the TWAS method, which is not only cheaper in practice but also portable, seem to be a suitable alternative. In addition to the corrosion increment analysis, another objective of this work was to study the microstructure and phase composition of the coatings for property comparison and a deeper understanding of corrosion kinetics.

Selected samples of the investigated materials were tested—uncoated steel, and coatings created using the HVOF method and TWAS method.

The results section describes the behaviours of the coatings after exposure to an environment of molten salt of 18% Na_2_SO_4_ and 82% Fe_2_(SO_4_)_3_ under cyclic loading to 690 °C. The coatings were studied using SEM/EDX cross-section analysis, X-ray diffraction, Raman spectroscopy, and corrosion kinetics.

Although the oxide layers formed on the coating surfaces are composed of different mixtures of oxides, the main oxides forming the most effective corrosion barrier are presented in all oxide layers—Cr_2_O_3_ and NiO. All corrosion oxide layers show the tendency to act as diffusion barriers for the inward penetration of the corrosive environment and prevent the oxidation of the whole coating.

In spite of the fact that the TWAS coatings generally exhibit significantly higher porosity compared to the HVOF coatings, selected TWAS coatings achieved comparable results to the HVOF coatings with the same coating thicknesses in the high-temperature corrosion test, and coatings formed using the TWAS method are recommended for application to the inner surface of coal-fired boilers.

## Figures and Tables

**Figure 1 materials-16-01712-f001:**
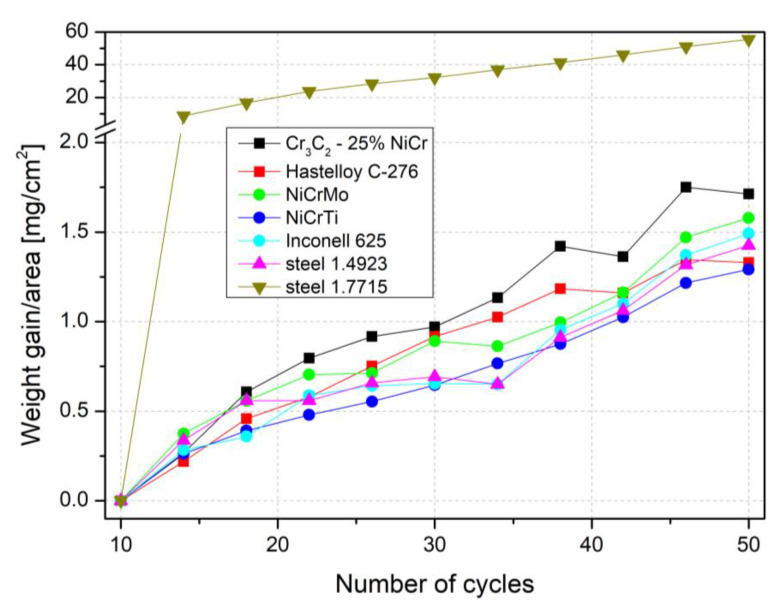
Graph for weight gain versus the number of test cycles for tested samples after exposure to the corrosion environment.

**Figure 2 materials-16-01712-f002:**
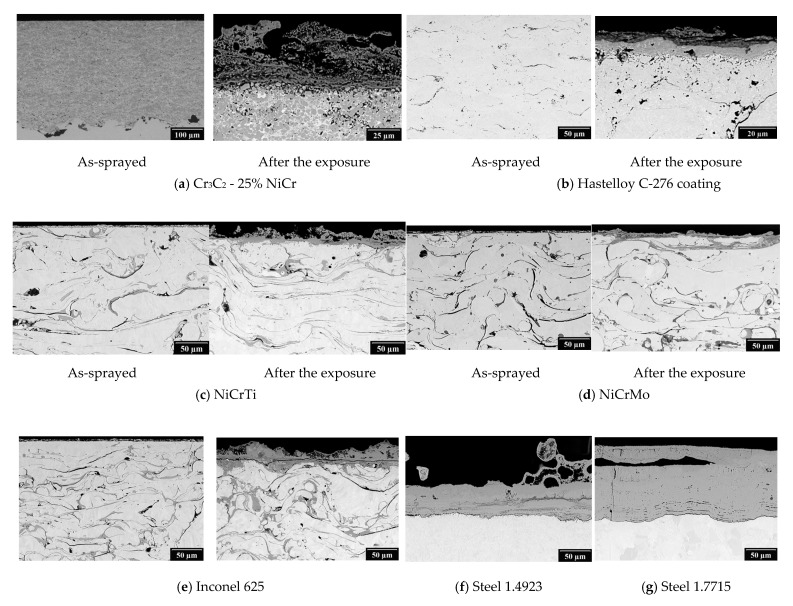
SEM photograph of the cross-section of tested samples.

**Figure 3 materials-16-01712-f003:**
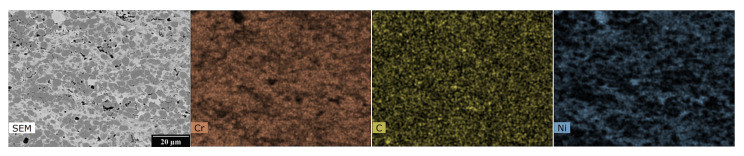
SEM photograph and EDX map of the cross-section of HVOF sprayed Cr_3_C_2_ - 25% NiCr coating—as-sprayed.

**Figure 4 materials-16-01712-f004:**
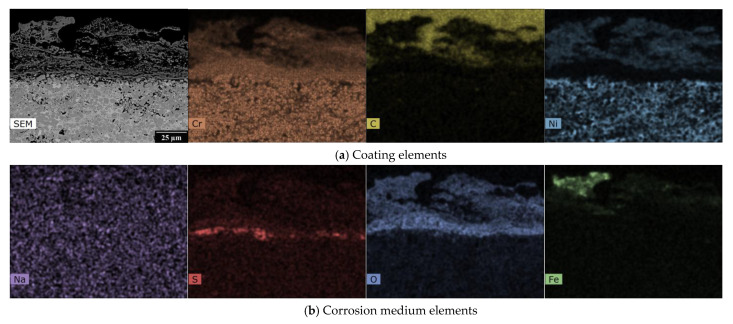
SEM photograph and EDX map of the cross-section of HVOF sprayed Cr_3_C_2_ - 25% NiCr coating after the exposure to the 18% Na_2_SO_4_ + 82% Fe_2_(SO_4_)_3_ corrosion environment.

**Figure 5 materials-16-01712-f005:**
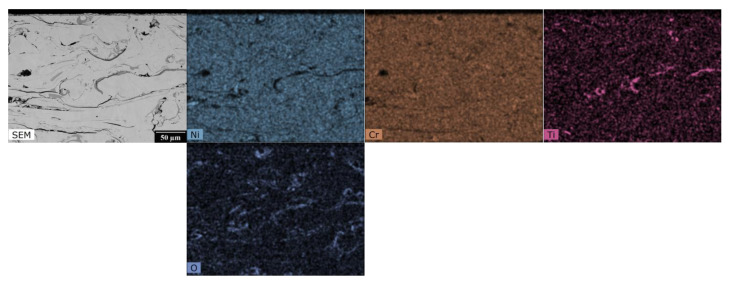
SEM photograph and EDX map of the cross-section of TWAS sprayed NiCrTi coating—as-sprayed.

**Figure 6 materials-16-01712-f006:**
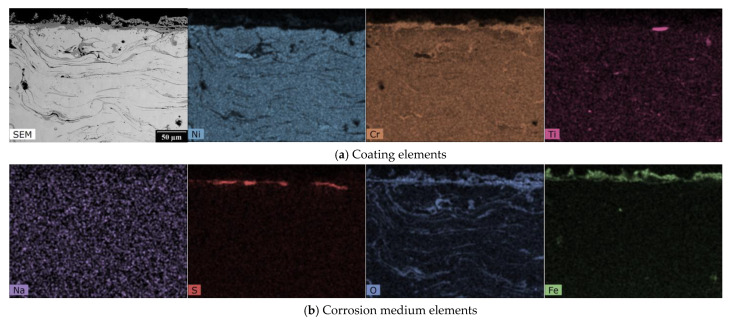
SEM photograph and EDX map of the cross-section of TWAS sprayed NiCrTi coating after the exposure to the 18% Na_2_SO_4_ + 82% Fe_2_(SO_4_)_3_ corrosion environment.

**Figure 7 materials-16-01712-f007:**
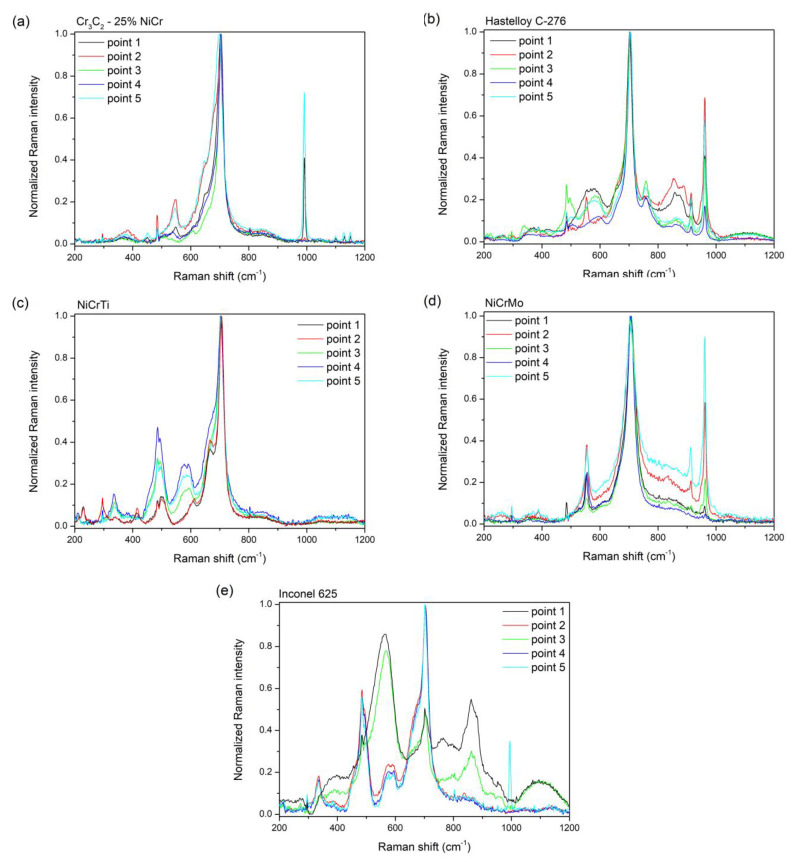
Measured Raman spectra from five independent positions on the samples (**a**) Cr_3_C_2_ - 25% NiCr, (**b**) Hastelloy C-275, (**c**) NiCrTi, (**d**) NiCrMo, and (**e**) Inconel 625.

**Table 1 materials-16-01712-t001:** Tested coatings.

Coating	Feedstock	Coating Technology	Form
Cr_3_C_2_ - 25% NiCr	Höganäs–Amperit 588.074	HVOF	Powder −45 + 15 µm
Hastelloy C-276	FST–M-341.33	HVOF	Powder −53 + 20 µm
NiCrTi	Taffa–45CT	TWAS	Wire 1.6 mm
NiCrMo	Oerlikon–Metco 8453	TWAS	Cored Wire 1.6 mm
Inconel 625	Oerlikon–Metco 8625	TWAS	Wire 1.6 mm

**Table 2 materials-16-01712-t002:** Chemical element composition of coatings (wt. %).

Coating	Ni	Cr	Ti	C	Al	Si	Fe	Mo	Nb	Ta	W	Other
Cr_3_C_2_ - 25% NiCr	18–22	Bal.		9–11			<0.5					
Hastelloy C-276	Bal	15.5					4	16			4	
NiCrTi	Bal.	45	0.3–1									0.5–1.85
NiCrMo–sheath	56.61	24.77		10.47	0.79	2.49	2.16	2.73				
NiCrMo–core	13.13	0.14	0.60		9.22	35.64	1.87	26.66				
Inconel 625	Bal.	21.5					2.5	9	0.1–3.5	0.1–3.5		

**Table 3 materials-16-01712-t003:** Chemical element composition of the steels (wt. %).

Steel	C	Si	Mn	P	S	Cr	Mo	V	Al	Sn	Ni	Nb
1.7715	0.1–0.18	<0.4	0.4–0.7	<0.025	<0.015	0.3–0.6	0.5–0.7	0.22–0.28	<0.02	<0.025		
1.4903	0.08–0.12	0.2–0.5	0.3–0.6	<0.02	<0.01	8.0–9.5	0.85–1.05	0.18–0.25			<0.4	0.06–0.1

**Table 4 materials-16-01712-t004:** Amount of corrosion products on the surface of tested samples after exposure to the corrosion environment taken using XRD analyses.

Coating	Corrosion Product [wt. %]								
	Oxide Spinel *	Cr_2_O_3_	Fe_2_O_3_	NiO	Ni	NiMoO_4_	m-FeMoO_4_	Fe_2_W	Na_2_SO_4_
Cr_3_C_2_ - 25% NiCr	47	35	12	7					
Hastelloy C-276	16	20	2	38		12	13		
NiCrTi	31	27	22	10					10
NiCrMo	30	21	12	13	16			6	2
Inconel 625	23	7	4	65					2
Steel 1.4923	3		84						13
Steel 1.7715	74		26						

* The structure of the spinel group of crystals is cubic and possesses the greatest possible number of symmetries. The formula is given as AB_2_O_4_, where A is divalent metal and B is trivalent metal [17]. In the case of bare steel samples, the designation oxide spinel hides magnetite (Fe_3_O_4_), which is consistent with the corrosion mechanism.

**Table 5 materials-16-01712-t005:** Presence of corrosion products on the surfaces of tested samples after exposure to the corrosion environment detected by Raman spectroscopy (vibration positions were compared with [22,23,24,25,26,27,28,29,30,31,32,33,34,35,36,37,38,39,40,41,42,43,44,45,46,47,48,49,50,51]).

Coating	Corrosion Product							
	Cr_2_O_3_	Fe_2_O_3_	NiFe_2_O_4_	FeCr_2_O_4_	NiMoO_4_	xMoO_4_	Fe_2_W	Na_2_SO_4_
Cr_3_C_2_ - 25% NiCr	x	x	x					x
Hastelloy C-276	x	x	x	x	x	x		
NiCrTi	x	x	x	x				
NiCrMo	x	x	x		x	x		
Inconel 625	x	x	x	x				x

## Data Availability

Data are contained within the article or are available on request from the corresponding author.

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
