# Peer review of "Hot Corrosion Behavior of TWAS and HVOF NiCr-Based Coatings in Molten Salt"

_materials, 2023, doi:10.3390/ma16041712_

Round 1

Reviewer 1 Report

Dear authors,

although your paper brings interesting results and contribution, the presentation is not adequate. Below some comments by section:

Abstract:
the steels used are not mentioned, only it was said "two uncoated materials". Avoid "more or less"

Introduction:
Before "For this reason" in the second paragraph, there are not references. Please bring references that support the background explained, also in other paragraphs.
In the second paragraph is the first time citing superalloys, thus it not make sense to use "These". Suggestion: These alloys, named superalloys, designed...
In the last paragraph: "NiCr-based materials" or NiCr-based coatings?
avoid expressions that are not informative like "observed properties". It is better to mention which properties

Materials and Methods

the chemical composition of the steels should also appear at Table 2.

In table 2 and/or in the text it is important also to make clear that the compositions are the nominal ones and not measured if I understood

How the mass variation was determined? Balance? which precision? In the results it was mentioned thermogravimetric test, but it seems that it was not the case, as it implies continuous mass measurement.

Results and Discussion:

In this section, the results are presented almost without discussion and not comparing with literature in the topics 3.1, 3.2, 3.3 and 3.4 (exception for the papers related to Raman phases). After, in the topic 3.5 some kind of discussion is presented. My suggestion is separate Results and Discussion as two sections.

It is strange to start in the cycle number 10 and in this cycle the weight gain be zero and the explanation is presented only after

Figures are not well described mainly regarding Figures 3 to 6.

The quality of the figures should be improved removing SEM bar and using only a more visible scale

It makes any sense to determine by Rietveld the amount of corrosion products? The intensity of outer layers will be always higher than the inner ones. The thickness evaluated will represent a different of fraction of the corrosion product layer for each alloy/coating

more than half of the references cited were used to identify Raman peaks

Finally, in general, the results were not well discussed and compared with the literature. The mass variaton, for example, is not compared with other studies. Low number of similar papers were cited.

Reviewer 2 Report

The Article “Hot corrosion behavior of TWAS and HVOF NiCr-based coatings in molten salt” by Kateřina Lencová et al. is devoted to applying of HVOF and TWAS methods for the obtaining of anti-corrosion coatings. The authors describe complex anti-corrosion 50-cyclic studies in molten salts of five sprayed coating samples as well as two samples of industrial steels. The results of the study will be useful to materials scientists and applicable in the metallurgical industry. The article, of course, should be published after major revision.

The Abstract needs great improvement. The abstract should be brevity, clarity, generality, self-contained. The innovation points are not clearly stated. Moreover, the significance of this study should be mentioned.

References. The bibliography is very poorly formatted.

The Introduction is not sufficient. Very few references are used in the text. References are poorly formatted. Comparison of thermal spraying methods, their advantages and disadvantages should be described above. In the text, it is necessary to highlight the novelty and feature of the study.

On Fig. 2b, there is no marker on the SEM image.

The presented EDX map is very hard to understanding. It would be more convenient to highlight the element with a color. At the moment it is not very clear, the selected elements for mapping are light areas? Why, for example, for sodium there is not even a moiré pattern, and sulfur is distributed only at the boundary of the layers? EDX images need to be presented in a more convenient form.

Low quality of Fig. 7.

Reviewer 3 Report

This paper reports the stability of various coatings applied to solid fuel boiler steels and exposed to a typical molten sulphate environment at 690°C. The most interesting part of this work is related to the demonstration that twin wire arc spray (TWAS) technology  can be used to replace the conventional and most expensive HVOF spraying method for the intended coating application. However, I have found some weak points in this work that need to be addressed before possible publication. Specific comments follow:

1.    Introduction: some more detailed information taken from literature with  proper references should be added in the introduction to justify the claimed statetements about the possible advantages in using the TWAS technology (e.g, mobility , costs, and so forth) in comparison to HVOF. 

2. Introduction: TWAS, HVOF: technical abbreviation terms should be defined the first time they are mentioned. 

3. Materials and Methods: if the main purpose of this work is to demonstrate that TWAS can be used to replace HVOF , it is not clear to me why the authors did not choose the same coatings to make the comparison between the two spray techniques. Using different coatings makes it more difficult to understand the real benefit of TWAS in comparison to HVOF. 

4. The authors used the Topas 5 software based on the Rietveld fitting method for the XRD quantitative analysis. Is it possible to add some more information on the strategy used by the authors to optimize the fitting parameters and thus to obtain reliable results in a such complex multiphase situation? 

5. par 3.3 : XRD analysis; 4th line: "means" and not "mean".

6. later in the same paragraph, there is a typo : "Cr2O3, Fe2O3 and Nio" in place of "Cr2O3, Fe2O3 a NiO".

Reviewer 4 Report

The paper reports the results of a comparative experimental investigation on HVOF and TWAS protective coatings for use at high temperature in coal-fired power plants.

 The subject is relevant, experimental activities were well conceived the manuscript is quite well written. Results appear reliable, discussion sufficiently complete, conclusions consistent.

 The following comments need to be addressed:

 Introduction:

- Please remove the first generic and unprecise sentence “Coatings… environment”

- par 2: please define acronym TWAS

- par. 2: TWAS is an alternative technique rather than a complementary one

- par. 3: replace mobile with portable

 2.2 Methods

- par.1 please either provide specific reference for the use of the selected salts combination for the simulation of coal fired power plants, if any, or modify the text

 3.1 Corrosion kinetics

- last par: please explain “during the first 10 cycles… evaluation”, not clear. Or remove

 3.2 SEM/EDX

- par. 1: it is not correct to define “protective” oxide scale at this moment. Replace with “formation of oxide scale”

- Fig.3-6: please provide a comment on these figures, otherwise unnecessary

 3.3. XRD analyses

- Table 3: a precision of XRD quantitative analysis of .1 % is absolutely not credible, please reduce to integer figures

- Table 3: the amount of residual Na2SO4 is surprisingly low. How were samples treated/cleaned before EDS/XRD?

 3.6 Economical aspects:

- par. 2: any reference available on average deposition efficiency of HVOF and TWAS?

 Conclusions

-par. 1 replace mobile with portable

Round 2

Reviewer 1 Report

The paper presents a considerable amount of results and characterization. The reviewed version improved the presentation but I still have some considerations:

About references, for me is strange that in the first paragraph in the Introduction section does not present any reference. Furthermore, some references are presented using [] and others (), please correct it.

Please detail the method used for chemical analysis in the cored wire.

The term thermogravimetric was removed from results but it is still in the conclusion, please remove it.

I do not think it is important to detail the brand and model of the scale used to weighing, but the information about precision of 0.1 mg should be improved.

I consider that in discussion, the values obtained for example in the mass variation should be compared with the literature.

The conclusion section is a bit longer than usual. It brings a lot of context, methodology and could focus more in the main results and discussion, showing the contribution from specific to general. Furthermore, I consider that is not common to do reference to any section as in "(see 3.1 Corrosion kinetics)"

Reviewer 2 Report

The Authors made corrections according to some comments (added markers, improved the quality of the figures).

However, the bibliography and references in the text are still poorly formatted. The Introduction is also uninformative.

For some reason, the formula of hematite is given and magnetite is not given when describing XRD analysis.

Reviewer 3 Report

The authors have provide satisfactory responses to all the enquiries of the reviewers. The manuscript is now acceptable for publication . 

Author Response

Let me thank you for your comments and recommendations. All of them were useful and will help to improve the quality of our article.